# Evaluation of all-for-one tourism development level: Evidence from Xinjiang production and construction corps, China

Yingyin Cui[1], Chunxiang Zhang[1]*, Bin Jiang, Ziwei Qin[1], Zhennan Liu[1], Yiwan Yang[2]

1 College of Science, Shihezi University, Shihezi, China, 2 College of Culture and Media, Xinjiang University of Science and Technology, Korla, China

* zhangchunxiang@shzu.edu.cn

## Abstract

All-for-one tourism represents a pivotal strategy to facilitate the transformation and enhancement of the tourism sector and promote the coordinated development of the economy and society within the contemporary context of China. Since its introduction by the Ministry of Culture and Tourism in 2016, governments at all levels have actively promoted the development of all-for-one tourism with remarkable results. Xinjiang Production and Construction Corps (XPCC), a unique governance system in Xinjiang, is rich in natural and cultural tourism resources and has achieved specific results in developing all-for-one tourism. Nevertheless, it also faces the outstanding problems of a low level of development and insufficient motivation. Therefore, this study focused on the XPCC as the research area and established an evaluation index system to assess the level of all-for-one tourism development across three dimensions: tourism potential, tourism benefit, and tourism format. Subsequently, a comprehensive analysis was conducted regarding the development level, spatial distribution characteristics, the coupling and coordination mechanisms among these dimensions, and the classification of tourism destinations. The results show that (1) XPCC's four major regions have unbalanced all-for-one tourism development, revealing disparities among divisions. (2) Spatial analysis shows that the Tianshan North Slope is advantageous, while southern Xinjiang and border areas are disadvantageous. (3) The coupling degree spans four stages, with significant differences in spatial distribution. (4) Tourism destinations can be categorized into three types: industrial integration-driven, economic and social-driven, and advantageous resource-driven. Accordingly, the optimization development strategies for all-for-one tourism in the XPCC were proposed: (1) integrating resources to optimize supply, (2) enhancing infrastructure to improve services, (3) deepening industrial integration, (4) strengthening brand building, (5) promoting regional cooperation. These strategies provide insights for optimizing all-for-one tourism development in similar regions.

## 1. Introduction

On May 17, 2024, the inaugural National Tourism Development Conference was convened. The meeting facilitated a comprehensive deployment and articulated specific requirements

**Data availability statement:** Data cannot be shared publicly because of the uniqueness of the governance system of the Xinjiang Production and Construction Corps. Data are available from the Xinjiang Production and Construction Corps Bureau of Statistics for researchers who meet the criteria for access to confidential data. Contact via Office Telephone: 0991-2890160 Address: No. 196 Guangming Road, Tianshan District, Urumqi City, Xinjiang Uyghur Autonomous Region Postal Code: 830002

**Funding:** A grant from the Social Science Foundation Project of the Xinjiang Production and Construction Corps. The project is titled 'Research on the Evaluation System and Cultivation Path of High-Quality Development of Red Tourism in Xinjiang Production and Construction Corps,' and it bears the project number 22YB08. (2) A grant from the Ministry of Culture and Tourism of China under the Social Science Research Project titled 'Research on the Patriotic Education Practice Path of Red Cultural Space in Shihezi, the First Military Reclamation City of the Republic.' The project number for this grant is 24DY37. the funders did play a supportive role in the review and editing stages of our manuscript

**Competing interests:** The authors have declared that no competing interests exist.

aimed at expediting the establishment of a robust tourism country and promoting the high-quality advancement of tourism. This initiative aligns closely with the overarching development concept of all-for-one tourism [1]. In June of the same year, during an investigative mission in Ningxia, China, the pertinent departments put forward strategies aimed at fostering a deeper integration between culture and tourism, alongside actively advancing the development of distinctive tourism and all-for-one tourism [2]. Since 2015, when the Ministry of Culture and Tourism issued the Notice on the Establishment of National All-for-One Tourism Demonstration Zones, 168 demonstration zones in China have successfully passed the acceptance testing process. Currently, the tourism industry is at a critical period of improving quality and efficiency. Therefore, the studies of all-for-one tourism are still of great significance in further clarifying the direction of development, optimizing resource allocation, improving tourism quality, promoting industrial advancement, and facilitating regional collaboration.

In 2008, the Shaoxing Municipal Government in Zhejiang Province, China, was the first to propose the development strategy of all-for-one tourism. Since then, the term 'all-for-one tourism' has been widely mentioned as a proprietary term in the development plans of diverse regions, and has become a hotspot of academic research in China. Early research focused on defining the concept of all-for-one tourism and constructing the theoretical framework [3,4]. As an academic term, Hu Xiaoran proposed that the essence of all-for-one tourism lies in the formation of clusters of tourism products or formats characterized by distinct features. This is achieved through the reintegration of resources within a spatial framework [5]. Lv Junfang proposed that all-for-one tourism is based on the contemporary notion of holistic development, transcending the limitations of traditional scenic spots. This approach encompasses various dimensions of tourism development, including regional construction, environmental protection, transportation, and catering services [6]. It aims to establish a scientifically optimized tourism system characterized by efficient resource allocation, organized spatial planning, diverse product offerings, and robust industry development, thereby facilitating the integrated development of tourism across the entire region. Additionally, as a conceptual term, Li Xinjian and colleagues have previously delineated the concept of 'all-for-one tourism'. They elucidate this concept through the framework of 'four new'(new resource view, product view, industry view, and market view) and 'eight complete' (whole factors, whole industry, whole process, all-round, whole time and space, whole society, whole departments, and whole tourists) [3]. In 2016, the Ministry of Culture and Tourism officially established the concept of all-for-one tourism as a widely recognized notion. Specifically, this concept entails designating tourism as the leading industry within a specific region, achieved through comprehensive and systematic enhancements and optimizations of economic and social resources. This includes but is not limited to tourism resources, associated industries, the ecological environment, public services, institutional frameworks, policies, regulations, and the overall quality of civilization. The objective is to achieve an organic integration of regional resources, foster integrated industrial development, promote social co-construction, and ensure sharing. Essentially, all-for-one tourism represents a novel concept and model for coordinated regional development, aiming to propel synchronized economic and social advancement through the medium of tourism [7]. In 2017, all-for-one tourism was incorporated into the government work report of the State Council, thereby establishing it as a national strategy. The scope of research pertaining to all-for-one tourism has broadened to include theoretical studies on development models [8,9], driving mechanisms [10,11], and influencing factors [12,13]. Furthermore, it encompasses practical explorations related to the establishment of all-for-one tourism demonstration zones and distinctive towns [14,15], the assessment of tourism competitiveness and development levels [16,17], industrial integration and coupling coordination [18,19], optimization of spatial characteristics and patterns [20], as well as initiatives for rural revitalization

and tourism-related poverty alleviation [21]. Though all-for-one tourism is China's original strategic concept and practice model, its core idea of emphasizing tourism development's comprehensive, holistic, and systematic nature has been widely resonated in the international academic community. Internationally related studies mainly include regional tourism planning [22,23], collaboration [24,25], sustainable tourism development [26], urbanization [27], and other fields. In summary, the breadth and depth of all-for-one tourism research continue to expand, and the theoretical system is becoming increasingly improved.

In recent years, the construction of model areas for all-for-one tourism has become increasingly mature. However, studies on experience summarization, effect evaluation, and model innovation must catch up. In the new era, how to further promote all-for-one tourism development has become a core issue requiring urgent attention. Evaluating the effects of a region's all-for-one tourism development is the key basis for optimizing development. A review of domestic and international studies shows that the all-for-one tourism development evaluation research scale is mainly based on national [20], provincial [13], and municipal factors [14]. The current research is devoted to comprehensively evaluating conditions and capacities for development, assessing development levels and efficiencies, and examining the impacts and effects of development, among other pertinent aspects. The research methodologies employed primarily consist of qualitative, quantitative, and case studies. Notably, the existing body of literature has overlooked the incorporation of tourism formats into a holistic evaluation framework, thereby neglecting the substantial influence of industrial integration on the progression of all-for-one tourism. Consequently, there is an urgent need to augment the evaluation system. Moreover, prior analyses have inadequately addressed regional differentiation characteristics, which are crucial for exploring development coordination and classifying tourism destinations. This limitation undermines the precision and efficacy of the strategic formulation. Additionally, there is a scarcity of studies focusing on the development of all-for-one tourism in the XPCC. In light of this, the present study focused on the XPCC as the research domain. It systematically constructed an evaluation index system for assessing the development level of all-for-one tourism, encompassing three pivotal dimensions: tourism potential, benefit, and format. Subsequently, the study measured and analyzed the all-for-one tourism development level within the XPCC and examined the coordination of development levels across these dimensions. Furthermore, it categorized the types of tourism destinations to unveil developmental disparities and inherent correlations among different divisions. Ultimately, this research endeavored to provide a scientific foundation for the optimal development of all-for-one tourism in the XPCC.

## 2. Materials and methods

### 2.1. Overview of the research area

The XPCC is situated in the Xinjiang Uyghur Autonomous Region (XUAR), located in northwest China. Its subordinate entities, including reclamation zones (each corresponding to a prefecture-level administrative division in Xinjiang), regiments (equivalent to township-level units), and enterprises (comparable to village-level units), are dispersed across various prefectures, cities, and counties within the XUAR. The XPCC implements a unique management system with a high degree of unity of the party, government, military, and enterprises, and undertakes the responsibility of garrisoning the frontier [28]. The XPCC has 14 divisions with a total area of about 70,600 km². According to the geographic location and the planning of XUAR, these 14 divisions belong to the Tianshan North Slope Development Zone (5th Division, 6th Division, 7th Division, 8th Division, 11th Division, 12th Division, 13th Division), the Tianshan South Slope Development Zone (1st Division, 2nd Division), the South Xinjiang

Development Zone (3rd Division, 14th Division), and the Border Port Economic Zone (4th Division, 9th Division, 10th Division) [29]. The 11th Division is dominated by the construction industry and has no direct connection with the tourism industry. Therefore, this study focused on the other 13 divisions as the research area (Fig 1).

During the 13th Five-Year Plan period in China, the XPCC comprehensively promoted the transformation of scenic spot tourism to all-for-one tourism. Among them, the 8th Division and 10th Division have successfully created national all-for-one tourism demonstration zones. During the 14th Five-Year Plan period, the XPCC will promote the strategy of all-for-one tourism in two ways. On the one hand, the existing demonstration zones will be helped to improve and upgrade their quality through refined management. On the other hand, potential divisions will be tapped to support the establishment of new demonstration zones. Therefore, the evaluation of the development level of all-for-one tourism will provide a scientific foundation for this measure.

## 2.2. Research method

### 2.2.1. EW-AHP combination weighting method.
In this study, the EW-AHP combination weighting method was employed to determine the weights of the indicators

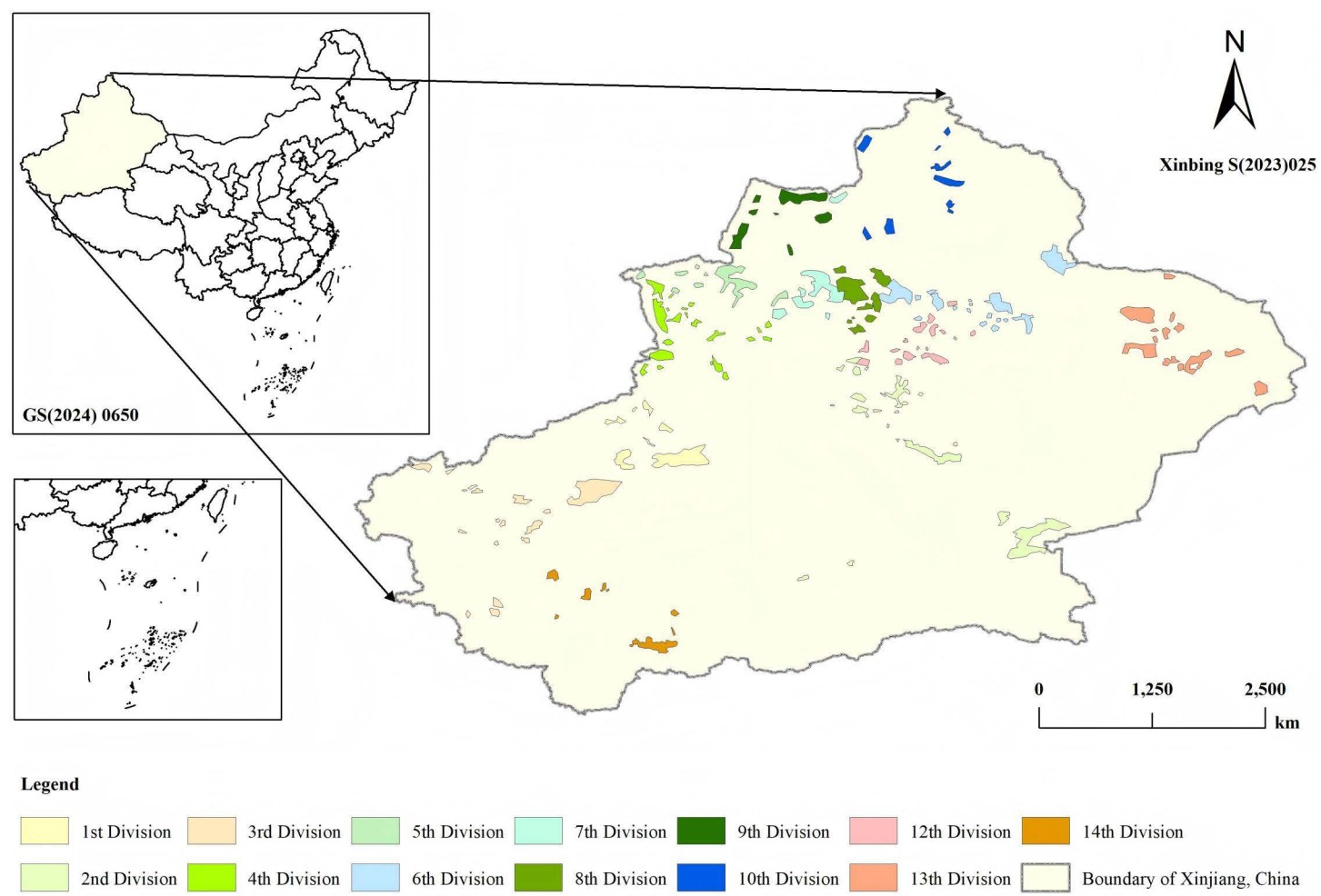

**Fig 1. Location of the XPCC within Xinjiang, China.** Note: The 11th Division of XPCC does not belong to the research area and is not shown in the figure.

within the evaluation system. This approach aims to mitigate the bias associated with objective assignment that arises from significant disparities in sample observations, at the same time, weaken the influence of subjective factors on the assignment of the hierarchical analysis method. The calculation procedures are based on related studies [30]. The final weights were calculated by the additive synthesis method, with weight coefficients set to 0.5, indicating that both objective and subjective weights are regarded as equally significant.

**2.2.2. Spatial correlation analysis.** Global spatial autocorrelation is used to describe the spatial distribution of attribute values throughout the research area and to determine whether elements are characterized by clustering in space. Afterward, local spatial autocorrelation is used to identify the agglomeration areas of high value, low value, and abnormal value in space and to reveal spatial heterogeneity [31]. This study used global spatial autocorrelation to test whether each dimension of XPCC's all-for-one tourism development is spatially correlated. Then, local spatial autocorrelation was used to analyze its correlation characteristics. Calculation steps refer to the related studies [32].

$$I = \frac{n\sum_{n}\sum_{n}^{i=1 j\neq i}w_{ij}(x_i - \overline{x})}{\sum_{n}\sum_{n}^{i=1 j\neq i}w_{ij}\sum_{n}^{i=1}(x_i - \overline{x})} \tag{1}$$

$$I_i = \frac{\sum_{n}^{j=1}w_{ij}(x_j - \overline{x})}{\sum_{n}^{i=1}(x_i - \overline{x})^2}\cdot(x_i - \overline{x}) \tag{2}$$

where: $n$ represents the overall count of entities in the study area, $w_{ij}$ is the spatial weight matrix constructed for analysis, $x_j$ symbolizes the attribute values of the $i$-th and $j$-th units, and $I_i$ is the local Moran's $I$ of the $i$-th observation point.

**2.2.3. Coupling coordination degree model.** The coupling coordination degree model serves as a tool for analyzing the extent of coordinated development among two or more systems. This model achieves this by calculating the degree of coupling and coordination, thereby illustrating the interdependence and constraints between the systems. Additionally, it elucidates the overall synergistic development of the system. The calculation steps refer to the relevant studies [33,34]. This research further evaluated the coupling coordination level of each dimension of the all-for-one tourism development in each division of the XPCC, utilizing the coupling coordination degree model.

$$C = \left(\frac{S_a \cdot S_b \cdot S_c}{\left[(S_a + S_b + S_c)/3\right]^3}\right)^{\frac{1}{3}} \tag{3}$$

$$T = \alpha S_a + \beta S_b + \gamma S_c \tag{4}$$

$$CD = (CT)^{\frac{1}{2}} \tag{5}$$

where: $C$ represents the coupling degree, $S_a$, $S_b$, and $S_c$ represent the evaluation index of tourism potential, tourism benefit, and tourism format subsystems, respectively, $T$ represents the comprehensive evaluation index, $CD$ represents the coupling coordination

degree, while α, β, and γ are the undetermined weight coefficients. Considering the centralized investigation of the three systems in this paper, the undetermined coefficients are assigned to 1/3.

**2.2.4. K-means clustering algorithm.** The k-means clustering algorithm is an unsupervised learning algorithm for dividing the dataset into K clusters. The optimal number of clusters is generally determined by the sum of the squares of the distances from the sample points within each cluster to the cluster's center of mass (SSE). The smaller the SSE is, the more convergent each cluster is. However, a smaller SSE is only sometimes preferable. In this study, the elbow method was used to determine the optimal number of clusters [35]. As the set number of clusters approaches the ideal value, SSE decreases rapidly, but beyond the ideal number of clusters, SSE decreases significantly slower. Therefore, the optimal number of clusters can be determined based on this rule.

## 2.3. Construction of evaluation index system

Based on international research studies [7,16,17] and the guidelines delineated in the 'Acceptance, Recognition, and Management Measures and Acceptance Criteria for All-for-One Tourism Demonstration Zones,' it is apparent that the evaluation index system for assessing the development level of all-for-one tourism encompasses multiple facets, namely resources, market dynamics, and economic benefits. In recent times, the significance of industrial integration in fostering the advancement of all-for-one tourism has gained prominence. Consequently, this study has incorporated the tourism format within the scope of the evaluation, adhering to the principles of systematicity, scientific rigor, and practical feasibility. Furthermore, an evaluation index system tailored to the XPCC's all-for-one tourism development level has been meticulously constructed.

Tourism potential, benefit, and format are intricately interconnected and mutually reinforcing elements that collectively determine the development trajectory of the XPCC's all-for-one tourism strategy. Tourism potential encompasses the essential prerequisites and inherent strengths that facilitate tourism growth, such as robust traffic infrastructure, abundant tourism resources, and high-quality tourism services. Conversely, tourism benefits signify the aggregate economic, social, and ecological advantages that stem from tourism endeavors. Tourism format, on the other hand, pertains to the operational and organizational modalities, encompassing the diversified and integrated pathways of development within the 'Tourism+' paradigm. Firstly, tourism potential acts as the cornerstone upon which the evolution of both tourism benefit and tourism format is built. Secondly, tourism benefits serve as the driving force and ultimate aspiration for bolstering tourism potential and refining the tourism format. Lastly, the tourism format is pivotal in harnessing potential and maximizing benefits. These three dimensions constitute the uppermost tier of the evaluation index system, encompassing 11 criterion layers and 20 specific indicators (Table 1).

## 2.4. Data sources

This study employed the most recently officially released data (i.e., data from 2022, given that 2023 data has yet to be published) to ensure a close alignment with the latest trends and characteristics of the tourism industry within the XPCC. The research drew upon a variety of authoritative sources, including the official website of the Ministry of Culture and Tourism of China, the 2022 Statistical Yearbook of XPCC, the 2022 Statistical Bulletin of National Economic and Social Development of the Divisions, the Annual Report on Government Information Openness, and other reliable statistical information available on the official websites of various governmental levels.

**Table 1. Evaluation index system for the development level of all-for-one tourism in the XPCC.**

| Target layer | System layer | Project layer | Indicator layer | Final weight |
|---|---|---|---|---|
| The all-for-one tourism development level of the XPCC | A1: Tourism potential (0.48) * | B1: Traffic condition (0.30) ** | C1: Annual Passenger Volume (10,000s) (0.32) *** | 0.046 |
| | | | C2: National and Provincial Highway Mileage (km) (0.27) *** | 0.039 |
| | | | C3: Number of airports (number) (0.41) *** | 0.059 |
| | | B2: Tourism resource (0.20) ** | C4: Number of Grade 3A and above attractions (number) (1.00) *** | 0.096 |
| | | B3: Tourism service (0.50) ** | C5: Number of travel agencies (number) (0.22) *** | 0.053 |
| | | | C6: Number of star-rated farmhouses (number) (0.22) *** | 0.053 |
| | | | C7: Number of star-rated hotels (number) (0.17) *** | 0.041 |
| | | | C8: Number of students majoring in tourism management in colleges and universities (number) (0.39) *** | 0.094 |
| | A2: Tourism benefit (0.20) * | B4: Economic benefit (0.49) ** | C9: Annual tourism revenue (billion yuan) (0.43) *** | 0.042 |
| | | | C10: Annual number of tourists (number) (0.34) *** | 0.033 |
| | | | C11: Annual per capita disposable income (yuan) (0.23) *** | 0.023 |
| | | B5: Social benefit (0.24) ** | C12: Proportion of employment in the tertiary industry (%) (0.48) *** | 0.023 |
| | | | C13: Urbanisation rate (%) (0.52) *** | 0.025 |
| | | B6: Ecological benefit (0.27) ** | C14: Percentage of forest cover (%) (0.53) *** | 0.029 |
| | | | C15: Proportion of good air quality days in a year (%) (0.47) *** | 0.025 |

*(Continued)*

**Table 1.** (Continued)

| Target layer | System layer | Project layer | Indicator layer | Final weight |
|---|---|---|---|---|
| | A3:<br>Tourism format<br>(0.32) * | B7:<br>Red tourism<br>(0.11) ** | C16:<br>Number of red tourism formats (number)<br>(1.00) *** | 0.035 |
| | | B8:<br>Rural tourism<br>(0.19) ** | C17:<br>Number of rural tourism formats (number)<br>(1.00) *** | 0.061 |
| | | B9:<br>Leisure tourism<br>(0.25) ** | C18:<br>Number of leisure tourism formats (number)<br>(1.00) *** | 0.080 |
| | | B10:<br>Industrial tourism<br>(0.23) ** | C19:<br>Number of industrial tourism formats (number)<br>(1.00) *** | 0.074 |
| | | B11:<br>Intangible cultural heritage tourism<br>(0.22) ** | C20:<br>Number of intangible cultural heritage tourism formats (number)<br>(1.00) *** | 0.070 |

Note: The symbols *, **, and ***represent the weights at the system, project, and indicator layers. The final weight of each indicator is calculated by multiplying these three weights.

## 3. Results and analysis

### 3.1. Measurement results of XPCC's all-for-one tourism development level

Using the weights assigned to each indicator, the development level of all-for-one tourism across the 13 divisions of the XPCC was evaluated, as presented in Table 2. At the regional level, the four major regions were ranked from highest to lowest in terms of all-for-one tourism development: the North Slope Development Zone (2.014) and the Border Port Economic Zone (0.985), followed by the Tianshan South Slope Development Zone (0.568), and finally the South Xinjiang Development Zone (0.253). A notable disparity in the development level of all-for-one tourism exists among these zones, indicating an overall unbalanced pattern of development within the XPCC. Taking into account each region's economic and social development, the Tianshan North Slope Development Zone, thanks to its relatively advanced economic and social infrastructure, has provided robust support for the thriving tourism industry. This has contributed to a positive trend of synergistic development between tourism and the local economy and society. Meanwhile, the Border Port Economic Zone, leveraging its unique border location and favorable ecological environment, offers distinctive conditions for specialized and differentiated tourism development. The Tianshan South Slope Development Zone boasts a national 5A-level tourist attraction and rich historical and cultural resources as the cornerstone for its tourism growth. However, a weak economy and inadequate infrastructure pose significant obstacles to further developing all-for-one tourism in this region. Conversely, the South Xinjiang Development Zone faces multiple challenges in advancing its all-for-one tourism due to its complex natural environment, remote geographical location, and relatively low level of economic and social development.

From the division level, according to the results of the all-for-one tourism level measurement, XPCC's divisions ranked from high to low in terms of the level of development of all-for-one tourism as follows: 8th Division (0.660), 4th Division (0.426), 1st Division (0.410), 6th Division (0.373), 10th Division (0.347), 7th Division (0.328), 12th Division (0.309), 13th Division (0.215), 9th Division (0.212), 3rd Division (0.184), 2nd Division (0.158), 5th Division (0.129), and 14th Division (0.069). The study used SPSS software to analyze the descriptive

**Table 2. Measurement results and rankings of the all-for-one tourism development level of XPCC.**

| Region | | Tourism potential | | Tourism benefit | | Tourism format | | Comprehensive level | | Regional comprehensive level | |
|---|---|---|---|---|---|---|---|---|---|---|---|---|
| | | Score | Ranking | Score | Ranking | Score | Ranking | Score | Ranking | Score | Ranking |
| Tianshan North Slope Development Zone | 5th Division | 0.042 | 11 | 0.079 | 10 | 0.008 | 12 | 0.129 | 12 | 2.014 | 1 |
| | 6th Division | 0.149 | 5 | 0.108 | 2 | 0.116 | 3 | 0.373 | 4 | | |
| | 7th Division | 0.206 | 4 | 0.102 | 7 | 0.020 | 10 | 0.328 | 6 | | |
| | 8th Division | 0.283 | 1 | 0.164 | 1 | 0.213 | 1 | 0.660 | 1 | | |
| | 12th Division | 0.108 | 8 | 0.103 | 5 | 0.098 | 4 | 0.309 | 7 | | |
| | 13th Division | 0.133 | 7 | 0.074 | 11 | 0.008 | 13 | 0.215 | 8 | | |
| Tianshan South Slope Development Zone | 1st Division | 0.213 | 3 | 0.105 | 4 | 0.092 | 5 | 0.410 | 3 | 0.568 | 3 |
| | 2nd Division | 0.033 | 13 | 0.095 | 8 | 0.030 | 7 | 0.158 | 11 | | |
| South Xinjiang Development Zone | 3rd Division | 0.098 | 9 | 0.056 | 12 | 0.030 | 8 | 0.184 | 10 | 0.253 | 4 |
| | 14th Division | 0.033 | 12 | 0.018 | 13 | 0.018 | 11 | 0.069 | 13 | | |
| Border Port Economic Zone | 4th Division | 0.135 | 6 | 0.084 | 9 | 0.207 | 2 | 0.426 | 2 | 0.985 | 2 |
| | 9th Division | 0.068 | 10 | 0.106 | 3 | 0.038 | 6 | 0.212 | 9 | | |
| | 10th Division | 0.220 | 2 | 0.103 | 6 | 0.024 | 9 | 0.347 | 5 | | |

statistics of the comprehensive level of the divisions. It can be seen that the distribution range is 0.069 to 0.660, and the gap is significant. The skewness value of 0.834 indicates that the comprehensive level has a clear positive skewness feature. The distribution of the dataset is shifted to the right, with a few extremely high values. At the same time, most of them are relatively concentrated in the lower range, highlighting the imbalance of XPCC's all-for-one tourism development. The kurtosis value is 1.120, indicating that the comprehensive level shows a spiky distribution pattern. The data points in the low-value areas are relatively concentrated. In contrast, the high-value areas are relatively sparse, confirming the unbalanced development of XPCC's all-for-one tourism and the low overall level. The kurtosis value is 1.120, indicating that the comprehensive level shows a spiky distribution pattern. The data points in the low-value areas are relatively concentrated. In contrast, the high-value areas are relatively sparse, further confirming the unbalanced development of XPCC's all-for-one tourism and the low overall level.

## 3.2. Spatial pattern characteristics of the level of each dimension of XPCC's all-for-one tourism

To delve deeper into the spatial distribution disparities across various facets of XPCC's all-for-one tourism, this study employed ArcGIS and Geoda software to conduct a spatial autocorrelation analysis of tourism potential, tourism benefits, and tourism formats within each division of XPCC. The Moran's $I$ index was calculated and subjected to rigorous testing. Notably, the Moran's $I$ index for each dimension exhibited a positive value. It surpassed the significance threshold at the $P$-value of 0.001, indicating a pronounced spatial agglomeration effect within each dimension of XPCC's all-for-one tourism (Table 3). Given the limitations of global spatial autocorrelation analysis in pinpointing the precise location of agglomeration areas, we further conducted a local spatial autocorrelation analysis to gain insights into the local spatial correlations across different dimensions of XPCC's all-for-one tourism. By consulting the local LISA agglomeration map, the spatial distribution of each dimension can be delineated into four primary zones: high-high agglomeration, high-low agglomeration, low-low agglomeration, and non-significant zones (Fig 2).

**3.2.1. Tourism potential dimension.** The high-high aggregation zones are mainly distributed in and around the core hub cities of the Tianshan North Slope Development Zone, Tianshan South Slope Development Zone, and Border Port Economic Zone, such as Alar City (seat of 1st Division Government), Shihezi City (seat of 8th Division Government), Beitun City (seat of 10th Division Government), and so on. These XPCC tourism node cities show excellent tourism potential due to their outstanding location advantages, good resource endowment conditions, and strong policy support. The high-low aggregation zones are distributed in the Tianshan North Slope Development Zone. There is a significant polarization effect in the region, which is manifested in its high tourism potential. At the same time, the surrounding areas are relatively low, and its radiation-driven role needs to be strengthened. The low-low agglomeration zones are mainly distributed in the South Xinjiang Development Zone, which is limited by multiple factors such as traffic accessibility, service

Table 3. Moran's $I$ test results for the XPCC's all-for-one tourism development level across different dimensions.

| Dimension | Moran's $I$ | $P$ value | $Z$ score |
|---|---|---|---|
| Tourism potential | 0.461 | 0.001 | 7.043 |
| Tourism benefit | 0.224 | 0.001 | 6.096 |
| Tourism format | 0.563 | 0.001 | 7.068 |

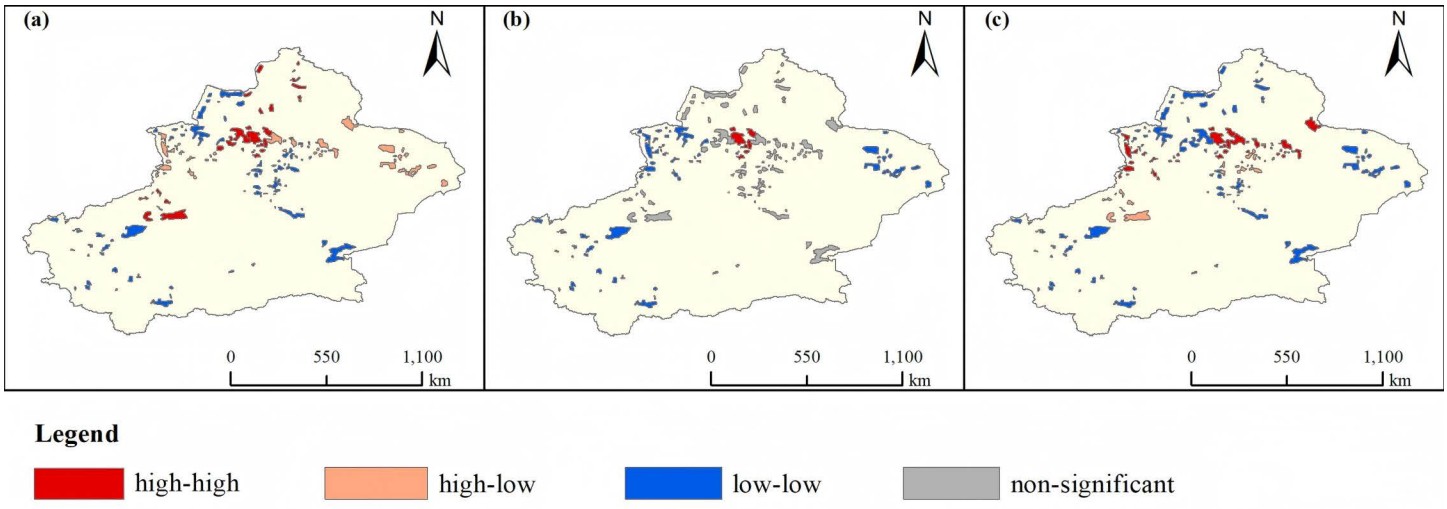

**Fig 2. Local autocorrelation LISA clustering for each dimension level of all-for-one tourism in the XPCC:** (a) Tourism potential, (b) Tourism benefit, (c) Tourism format.

facilities construction, and resource development. As a result, the potential for tourism has yet to be fully realized.

**3.2.2. Tourism benefit dimension.** The high-high aggregation zone is distributed in the 8th Division of the Tianshan North Slope Development Zone. The economic, social, and ecological benefits in the region show a promising trend of complementing each other. The low-low agglomeration zones are mainly concentrated in the South Xinjiang Development Zone and in some divisions (3rd Division, 4th Division, 5th Division, 13th Division, and 14th Division) of the Border Port Economic Zone and Tianshan North Slope Development Zone. These areas exhibit a lagging performance in tourism development, as evidenced by a notable disparity in the tourism economy's input-output ratio, thereby hindering the full potential of these regions to contribute comprehensively to the local economy and society. Despite the remarkable tourism benefits observed in the high-aggregation zones, the spatial dissemination of these positive effects is hampered by various factors, making it challenging to disseminate these benefits to adjacent regions effectively. Consequently, a substantial number of non-significant zones have emerged, which not only underscores the uneven spatial distribution of tourism benefits but also emphasizes the pressing need for fostering coordinated tourism development across different regions. To mitigate this issue, it is imperative to break down spatial barriers, facilitate the sharing of resources, and harness complementary advantages, ultimately aiming to enhance the overall performance of the tourism industry.

**3.2.3. Tourism format dimension.** The high-high aggregation zones are predominantly located within specific divisions (the 4th, 6th, and 8th Divisions) of the Tianshan North Slope Development Zone and the Border Port Economic Zone. The high-high aggregation zones are predominantly located within specific divisions (the 4th, 6th, and 8th Divisions) of the Tianshan North Slope Development Zone and the Border Port Economic Zone. These zones exhibit a promising trend towards the deep integration of the tourism industry with diversified economic sectors. In stark contrast, the high-low aggregation zones are scattered across selected divisions of both the Tianshan North Slope and South Slope Development Zones (1st and 12th Divisions). Notably, the 1st Division stands out for its distinctive red tourism offerings, while the 12th Division boasts a well-rounded portfolio of tourism formats.

Compared to their neighboring regions, these two divisions possess remarkable competitive edges. Conversely, the low-low aggregation zones are prevalent in the Tianshan North Slope Development Zone, Tianshan South Slope Development Zone, South Xinjiang Development Zone, and Border Port Economic Zone. This observation underscores the imbalance in the development of tourism formats across various regions, indicating ample opportunities for enhancement and refinement in differentiating and diversifying tourism offerings. This analysis underscores the critical need to address the disparities in tourism development across diverse regions. It emphasizes the importance of fostering coordinated tourism growth and capitalizing on the unique competitive strengths of each area to drive balanced and sustainable tourism development.

### 3.3. Tourism potential-tourism benefit-tourism format coupling coordination degree

Because spatial correlation analysis falls short in fully elucidating the combined impact of the three dimensions, namely tourism potential, tourism benefit, and tourism format, on overall coordinated development, this study introduces the coupling coordination degree model. The primary objective of this model is to rigorously examine the intrinsic correlation mechanism among these dimensions and to provide a comprehensive evaluation of the coupling coordination degree in the context of all-for-one tourism development across various divisions within the XPCC.

Tourism potential is the basis for carrying out tourism activities. Tourism benefit is the standard to measure the economic, social, and ecological impacts. In addition, the tourism format is the driving force behind the promotion of industrial transformation and upgrading. Therefore, the coupling coordination degree of each division in the three dimensions was calculated. Based on the coupling coordination degree level division standard proposed by related research (Table 4) [36,37], XPCC's divisions were classified into four levels: extremely uncoordinated, close to uncoordinated, barely coordinated, and primarily coordinated, which were visualized with the help of ArcGIS software (Fig 3).

Firstly, the 2nd, 5th, 13th, and 14th Divisions are categorized into extremely uncoordinated types. These divisions exhibit a notable lack of synergy between tourism potential, tourism benefit, and tourism format, ultimately leading to suboptimal levels of coordinated development. Specifically, the 4th Division faces challenges such as remoteness, inadequate tourism service infrastructure, and limited resource development, which impede the growth of tourism

**Table 4. Gradation criterion of coupling coordination degree.**

| Range of coupling coordination degree | Grade |
| --- | --- |
| [0.0, 0.1] | Extremely uncoordinated |
| [0.1, 0.2] | Serious uncoordinated |
| [0.2, 0.3] | Moderate uncoordinated |
| [0.3, 0.4] | Mild uncoordinated |
| [0.4, 0.5] | Close to uncoordinated |
| [0.5, 0.6] | Barely coordinated |
| [0.6, 0.7] | Primary coordinated |
| [0.7, 0.8] | Intermediate coordinated |
| [0.8, 0.9] | Good coordinated |
| [0.9, 1.0] | High quality coordinated |

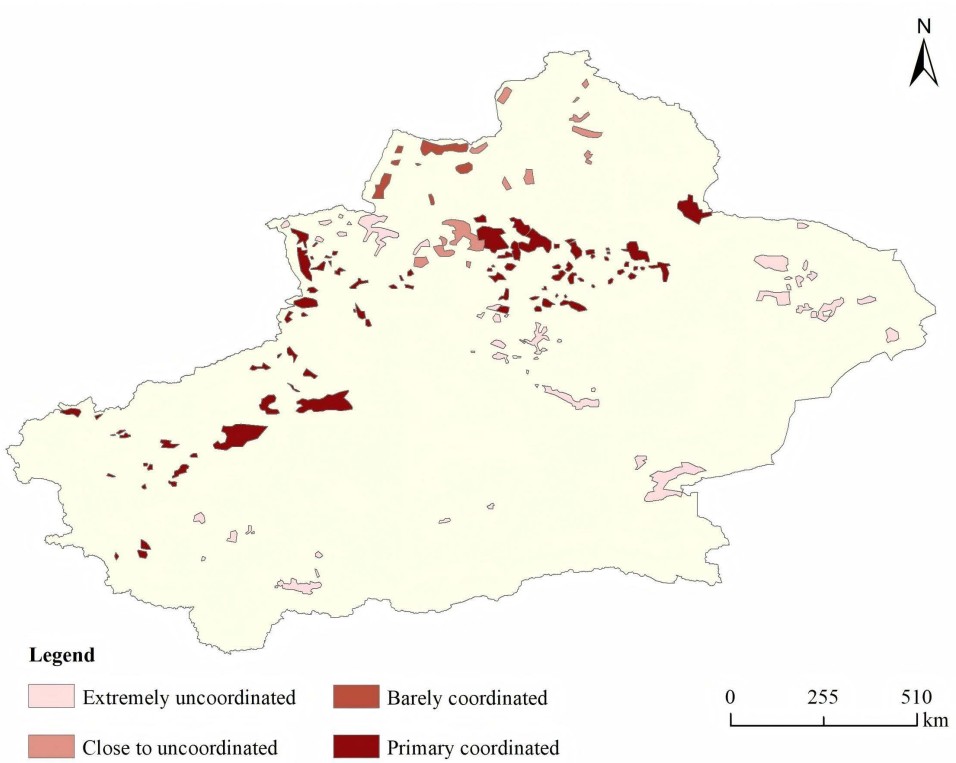

**Fig 3. Spatial distribution of coupling coordination degree of tourism potential-tourism benefit-tourism format.**

benefits and the diversification of the tourism industry, thereby preventing the formation of a positive feedback loop among these components.

Secondly, the 7th and 10th divisions are identified as close to uncoordinated. In these divisions, the interaction between tourism potential, benefit, and format is limited and unstable. Taking the 10th Division as an illustrative case, its heavy reliance on a singular tourism format, primarily red and rural tourism, results in rapid depletion of tourism potential and sluggish growth in benefits, thereby intensifying the risk of developmental imbalance.

Thirdly, the 9th Division is classified as barely coordinated. This division has established a relatively stable interaction among tourism potential, benefit, and format. Notably, the ecological benefits of the 9th Division are substantial, providing a solid foundation for tourism resource development and favorable conditions for the expansion of tourism formats. However, the depth and breadth of this interaction are insufficient, hindering the maximization of the synergistic effect.

Fourthly, the primary coordinated type encompasses the 1st, 3rd, 4th, 6th, 8th, and 12th Divisions. These divisions have established significant interaction mechanisms among tourism potential, benefit, and format, achieving preliminary coordinated development. The 8th Division exemplifies this through precise positioning of tourism resources, effective integration of regional advantages, leveraging red tourism as a core strength, and innovating and diversifying tourism formats such as rural, industrial, leisure, and intangible cultural heritage tourism. This strategy not only intensely activates tourism potential but also ensures continuous growth in tourism benefits, serving as a benchmark for the coordinated development of all-for-one tourism within the XPCC. In anticipation of future changes and challenges within

the tourism market, it is crucial to continuously refine the interaction mechanisms among tourism potential, tourism benefit, and tourism format to ensure sustained progress and resilience in the sector.

Spatially, there exists a notable imbalance in the coordinated development of XPCC's all-for-one tourism. Zones characterized by extreme incoordination exhibit distinct dispersion patterns, encompassing the Tianshan North Slope Development Zone, the Tianshan South Slope Development Zone, the South Xinjiang Development Zone, and the Border Port Economic Zone. Adjacent to these extremely uncoordinated zones are primarily areas that lie between them and the coordinated zones, demonstrating a gradual transition from uncoordinated to coordinated development. Meanwhile, barely coordinated zones are situated within the Border Port Economic Zone, possessing a certain foundation for coordinated development owing to their unique geographical advantages. Primary coordinated zones exhibit relatively concentrated developmental trends across the four major zones and exert a certain leading and exemplary influence on neighboring regions.

### 3.4. Types of all-for-one tourism destinations in the XPCC

While the coupling coordination degree analysis offers valuable insights into the overall coordination status of tourism development, it falls short in elucidating the specific characteristics and strengths of individual divisions within this sphere. To address this limitation and facilitate the formulation of targeted optimization strategies, it is imperative to further categorize the types of tourism destinations. Drawing upon existing research findings [38–40], this study refines the guideline layer of XPCC's evaluation index system for the development level of all-for-one tourism into three core components: economic and social foundation (B1, B2, B4, B5), resource endowment (B2, B6), and industrial integration (B7, B8, B9, B10, B11). Subsequently, utilizing the K-means clustering algorithm, the 13 divisions of XPCC were segmented into distinct types of all-for-one tourism destinations. The optimal number of clusters, determined through the elbow method, was found to be three (Fig 4). The classification outcomes are depicted in Fig 5. By aligning these classifications with the actual progress of all-for-one tourism in each division, the tourism destinations were categorized into three distinct types.

**3.4.1. Industrial integration-driven (4th Division, 6th Division, 8th Division, 12th Division).** The defining characteristic of this category is the deep integration of primary, secondary, and tertiary industries, leading to the creation of a unique and diversified tourism format system. Specifically, the 4th Division has emerged as a leading example of industrial tourism, leveraging national industrial tourism demonstration sites like the Ipal Khan Lavender Sightseeing Park. The 6th Division, on the other hand, is renowned for its Kazakh felt and cloth embroidery, with intangible cultural heritage tourism being its primary attraction. The 8th Division has successfully established the influential red tourism brand of 'The First City of Military Reclamation of the Republic.' While the 12th Division may not excel in rural, red, leisure, industrial, or other tourism formats, its diversified tourism offerings represent a unique developmental asset. Consequently, a critical and urgent issue for this type of destination is how to further deepen industrial integration to fully harness the intrinsic value of tourism resources and more effectively drive economic and social development.

**3.4.2. Economic and social-driven (1st Division, 3rd Division, 5th Division, 7th Division, 13th Division).** The prominent feature of this type is that regional economic and social development serves as the key driving force behind the prosperity of tourism. As economic strength increases, social infrastructure improves, and residents' living standards rise, the tourism industry has experienced rapid growth, gradually becoming an essential pillar of the local economy. The 1st and 3rd Divisions are the core development areas of the XPCC in southern Xinjiang, while the 5th, 7th, and 13th Divisions are located in the Tianshan North

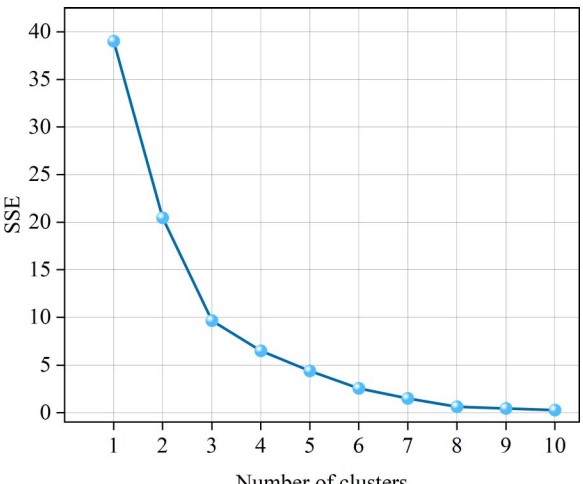

**Fig 4. SSE corresponding to different number of clusters.**

**Fig 5. 3D Scatter Plot of Clustering Results.**

Slope Development Zone. With relatively advantageous location conditions, the economic and social development level is relatively high, which has laid a solid foundation for tourism development. However, the in-depth exploration of resource endowments and the diversified development of products are insufficient. For example, the 7th Division has discovered the 'Gobi Mother' cultural tourism IP. However, sustainable development efforts are insufficient,

and tourism products' innovative design and marketing are lagging behind. Consequently, the primary challenge for the development of this type lies in improving economic and social benefits, deepening the careful exploration and development of characteristic resources, and enhancing tourism appeal and market competitiveness.

**3.4.3. Advantageous resource-driven (2nd Division, 9th Division, 10th Division, 14th Divisions).** The distinctive feature of this type of tourism development is its reliance on the region's unique endowment of natural and human resources to foster sustainable growth. The 2nd, 9th, and 10th Divisions boast suitable ecological environments and abundant natural resources. The 9th Division has the renowned Xiaobaiyang Sentry Post, while the 10th Division features the first national 5A-level tourist attraction in the XPCC, significantly enhancing its tourism appeal. The 14th Division has tapped into the unique cultural tourism resources of 'veteran spirit,' which holds considerable potential for tourism development. However, shared challenges these areas face include remote locations, high transportation costs, and limited tourist appeal. To overcome these obstacles, it is essential to focus on improving traffic accessibility, optimizing the use of distinctive resources, boosting the popularity of tourism brands, and stimulating tourists' interest in visiting.

## 4. Conclusions

Drawing upon the three dimensions of tourism potential, tourism benefit, and tourism format, this study establishes a comprehensive evaluation index system tailored to assess the all-for-one tourism development level of the XPCC. From a holistic and multi-faceted perspective, this research conducted an in-depth analysis of the XPCC's all-for-one tourism development level. Furthermore, it delved into these three dimensions' coupling coordination. Building on this foundation, the K-means clustering analysis method was employed to categorize the XPCC's all-for-one tourism destinations in a rational manner. This classification not only offers a theoretical foundation but also provides practical guidance for optimizing and further developing the XPCC's all-for-one tourism industry.

The conclusions are as follows: (1) The XPCC's all-for-one tourism development shows a significant imbalance. At the regional level, the Tianshan North Slope Development Zone demonstrates superior development, followed by the Border Port Economic Zone. In contrast, the Tianshan South Slope Development Zone and the South Xinjiang Development Zone lag behind. At the divisional level, low values of all-for-one tourism development are concentrated in certain areas, with few high values present, indicating that the overall development level of XPCC's all-for-one tourism is low. (2) The various dimensions of XPCC's all-for-one tourism display a noticeable spatial aggregation effect. Areas with advantageous tourism potential, benefit, and format are mainly concentrated in the Tianshan North Slope Development Zone, while southern Xinjiang and some border areas form low-value clusters, underscoring the urgent need for regional coordination and differentiated development. (3) XPCC's all-for-one tourism reveals gradient differences in the coupling and coordination of tourism potential, benefit, and format, covering four levels ranging from extremely uncoordinated to primarily coordinated. Spatially, this is characterized by a discrete distribution of extremely uncoordinated zones and a centralized distribution of primarily coordinated zones. Regions with higher degrees of coordination benefit from superior location conditions and a solid foundation for tourism development. Conversely, areas with low coordination degrees face various constraints, including insufficient tourism resource development, remote geographical locations, and a lack of diverse tourism formats. (4) Based on different driving factors, XPCC's all-for-one tourism destinations can be classified into three categories: industrial integration-driven, economic and social-driven, and advantageous resource-driven.

## 5. Discussion

In the realm of research pertinent to evaluating tourism development levels, the evaluation index system for all-for-one tourism development formulated by Zeli Hu and colleagues encompasses three primary domains: industrial, spatial, and factorial [41]. This system emphasizes intricate power dynamics such as economic contributions, transportation infrastructure, and the demographic structure of the tourism industry. Compared with this framework, the present study incorporates not only hard power considerations but also underscores the significance of tourism services, thereby reflecting a concentration on tourism's soft power. Haijun Liu and colleagues, in constructing their evaluation index system, focused on three primary segments: tourism development, tourism resources, and tourism support [42]. While encompassing tourism resources and public facility resources, their system exhibits a slight limitation in terms of the diversity of tourism formats. In contrast, the index system developed in this study thoroughly considers the regional characteristics and resource advantages of XPCC, particularly accentuating red tourism, rural tourism, and other locally distinctive tourism forms to ensure the index system's targeted relevance. Lei Tian and colleagues formulated a tourism industry subsystem centered on industrial scale, structure, and development potential [43]. Within the tourism benefit subsystem of this study, economic benefits are taken into account, along with social and ecological benefits, thereby illustrating the multifaceted impact of all-for-one tourism development. In their assessment of quality tourism development in China, Hong He and colleagues constructed a multidimensional index system encompassing tourism economy scale, development speed, tourism supply quality, tourism industry benefits, and tourism industry structure [44]. Although this system delves deeply into the quality of tourism supply, it somewhat lacks diversity in tourism formats and the exploration of tourism potential. The present study integrates all pertinent indicators into tourism potential and tourism format, thereby ensuring the comprehensiveness of the index system while also highlighting the unique features and highlights of XPCC's comprehensive tourism.

In summary, the index system developed in this study builds upon previous multidimensional comprehensive evaluation frameworks, integrating the unique regional characteristics, resource endowments, and development realities of XPCC. It meticulously considers the profound impact of various tourism formats on the progression of all-for-one tourism. Additionally, this study conducts a thorough and insightful analysis from multiple perspectives, including spatial layout, coordinated development, and tourism destination classifications, thereby broadening the horizon for assessing the development level of all-for-one tourism. However, during the construction of the evaluation system in this study, there was a notable omission: the insufficient consideration of dynamic factors from the demand-side perspective, particularly tourist satisfaction. This limitation somewhat restricts the study's depth and scope. Future research endeavors could strive for a more systematic and profound exploration of all-for-one tourism development levels from the demand-side angle, aiming to unveil its internal rationale and external driving mechanisms comprehensively.

## 6. Suggestions

Based on the current situation of the XPCC's all-for-one tourism development, this study proposed the following optimization strategies.

Firstly, integrating tourism resources and optimizing product supply is paramount. Addressing the prevalent issues of small-scale, scattered distribution and limited economic returns in XPCC's tourism resource development, integrating these resources is a crucial solution. Notably, the 1st Division has achieved remarkable success by establishing the second national 5A-level tourist attraction through the amalgamation of the Three-Five-Nine Reclamation Memorial Hall, the

Taklimakan Desert Gate Scenic Area of the 11th Regiment, and various other resources, including red tourism, agricultural tourism, sports, and recreation. This achievement is a valuable benchmark for other divisions' resource integration efforts. Taking the 14th Division as a case in point, it can draw inspiration from this success by integrating the resources of the First Ranch, Pishan Farm, and the 47th Regiment. By deeply fusing the distinctive elements of red culture, Kunlun culture, ecological culture, agriculture and animal husbandry culture, and ethnic culture, the 14th Division can develop a more captivating and all-for-one tourism area. The strategy for all-for-one tourism product supply underscores the importance of 'full-time coverage,' aiming to activate the off-season market by creating seasonal tourism products, such as ice and snow tourism, ensuring continuous tourist activities throughout the year. Furthermore, by expanding tourism products into the nighttime economy, including night tours, night shopping, night performances, night entertainment, and night exhibitions, the temporal dimension of tourism products can be broadened, enriching the tourist experience and fostering product diversification.

Secondly, the enhancement of infrastructure and service quality is essential. Addressing the issue of transportation accessibility, particularly in remote areas such as the 9th and 14th Divisions, the development of general aviation should be actively pursued to bolster connectivity with key source markets and neighboring tourist destinations. Policy guidance and market incentives should be employed to facilitate the renovation and upgrading of existing service infrastructure to tackle the shortage and low quality of catering and lodging facilities, such as farmhouses and tourist hotels. Additionally, the optimal allocation of medium and high-end service offerings is crucial to meet the diverse and high-quality demands of the market. As the significance of smart tourism systems becomes increasingly prominent, it is recommended that the XPCC's smart tourism system be centered around the 'one center and two platforms' framework. This includes the tourism big data center, an innovative tourism management platform, and an intelligent tourism service and marketing platform. The aim is to facilitate government decision-making and supervision, support the digital operations of enterprises, enhance the service experience for tourists, and enrich scientific research data resources. Furthermore, the sustainable improvement of tourism service quality necessitates robust talent training. Leveraging the resources of XPCC's colleges and universities, particularly Shihezi University in the 8th Division and Tarim University in the 1st Division, is crucial. These institutions should systematically cultivate high-level management talents and grassroots tourism professionals, providing solid human resources support for tourism specialization and high-quality development.

Thirdly, industrial integration should be deepened, and the primary, second, and third industries should be linked. In the aspect of 'agriculture + tourism,' agricultural production, rural ecology, and farmers' lives should be transformed into tourism resources, with the creation of idyllic complexes, agricultural sightseeing parks, and rural lodgings, as well as the incorporation of modern agricultural science and technology elements, such as the display of intelligent agriculture and the experience of deep-processing of agricultural products, to enhance the sense of science and technology and interaction. In the aspect of 'industry + tourism,' industrial sites, modern factories, and industrial parks should be transformed into new tourism highlights. Developing industrial heritage tourism, preserving and transforming old factories and production lines to show the history of industrial civilization, promoting industrial experience tourism, opening high-tech production lines, and encouraging tourists to participate in the production of products to feel the charm of industry and technological innovation intuitively. In the aspect of 'education + tourism,' patriotic education bases can be built based on the historical evolution of XPCC, red cultural sites, and heroic deeds. Popular science education bases can be established by combining XPCC's distinctive advantages in culture, science and technology, ecology, and other fields. By entirely using the XPCC's natural resources, such as ice and snow, deserts, and grasslands, XPCC can develop outdoor sports projects such as hiking, skiing, and horseback riding and create

outdoor sports experience bases. Based on this, study tours can be actively developed to meet the demand for diversified educational experiences.

Fourthly, the construction of tourism brands is strengthened to increase market recognition and influence of XPCC's notable red tourism brand, particularly 'XPCC Memory'. This initiative includes the design and promotion of cartoon imagery and cultural products that reflect the unique characteristics of XPCC, thereby fostering brand recognition and affinity. By relying on the heroic deeds and garrison spirit, a series of cultural tourism intellectual properties has been created, including high-quality literary works, tourism performances, and IPs such as 'Borderland Populus' and 'Veterans of the Sea of Sand'. Moreover, various divisions have established distinct tourism brands based on their unique resource endowments and cultural significance. For instance, the 3rd Division has introduced 'Silk Road Tang King City - New Oasis of Reclamation,' while the 8th Division features 'The First City of Military Reclamation of the Republic.' To promote the development of these tourism brands, a 'five-in-one' publicity and marketing mechanism should be established, encompassing government support, department coordination, enterprise linkage, media follow-up, and tourist participation. This mechanism leverages new media platforms, such as short videos, live broadcasts, and mobile applications, in combination with film and television, events, and activity marketing. By integrating these strategies, XPCC aims to strengthen its tourism brands, ensuring that they resonate with both domestic and international tourists, thereby contributing to the sustainable development of the tourism industry in the region.

Fifthly, promoting regional cooperation and integrated development between the XPCC and XUAR should be prioritized. Based on the development level of all-for-one tourism across various divisions, three primary areas for tourism co-construction and resource sharing can be established. (1) The Tarim Cultural Tourism Circle leverages the Silk Road culture of southern Xinjiang. By connecting the 1st, 2nd, 3rd, and 14th Divisions through major transport routes such as G217, this area aims to create tourism products centered on land reclamation, historical significance, deserts, and oases. These products will showcase the region's unique features and attract tourists seeking a blend of cultural and natural experiences. (2) The Tianshan Ecological and Cultural Tourism Belt focuses on the ecology of the Tianshan Mountains and the culture of military reclamation. Combining the efforts of the 6th, 7th, 8th, 12th, and 13th Divisions, this belt aims to develop tourism products emphasizing ecology, red culture, leisure, and wellness. These products will provide tourists with opportunities to experience the natural beauty and cultural heritage of the Tianshan Mountains while promoting sustainable tourism practices. (3) The Western Border Cross-border Tourism Belt encompasses the 4th, 5th, 9th, and 10th Divisions. This area aims to build a cross-border tourism cooperation zone and promote the development of border tourism branding. By leveraging the region's unique geographical and cultural features, this belt will offer tourists a unique and immersive cross-border travel experience. At the same time, exploring the XPCC-XUAR, division-prefecture and city, and regiment-county and township multi-level regional tourism integration and development model, optimizing the allocation of resources, and promoting synergistic development.

## Author contributions

**Conceptualization:** Yingyin Cui.

**Data curation:** Bin Jiang.

**Formal analysis:** Yingyin Cui.

**Funding acquisition:** Chunxiang Zhang.

**Visualization:** Ziwei Qin, Zhennan Liu.

**Writing – original draft:** Yingyin Cui.

**Writing – review & editing:** Chunxiang Zhang, Yiwan Yang.

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
