## [Decision Letter · Decision Letter 0]

5 Nov 2024

PONE-D-24-46493Evaluation of All-for-One Tourism Development Level: Evidence from Xinjiang Production and Construction Corps, ChinaPLOS ONE

Dear Dr. Zhang,

Thank you for submitting your manuscript to PLOS ONE. After careful consideration, we feel that it has merit but does not fully meet PLOS ONE’s publication criteria as it currently stands. Therefore, we invite you to submit a revised version of the manuscript that addresses the points raised during the review process.

We look forward to receiving your revised manuscript.

Kind regards,

You-Yu Dai

Academic Editor

PLOS ONE

3. In this instance it seems there may be acceptable restrictions in place that prevent the public sharing of your minimal data. However, in line with our goal of ensuring long-term data availability to all interested researchers, PLOS’ Data Policy states that authors cannot be the sole named individuals responsible for ensuring data access (http://journals.plos.org/plosone/s/data-availability#loc-acceptable-data-sharing-methods).

5. We note that Figures 1, 2 and 3 in your submission contain [map/satellite] images which may be copyrighted. All PLOS content is published under the Creative Commons Attribution License (CC BY 4.0), which means that the manuscript, images, and Supporting Information files will be freely available online, and any third party is permitted to access, download, copy, distribute, and use these materials in any way, even commercially, with proper attribution. For these reasons, we cannot publish previously copyrighted maps or satellite images created using proprietary data, such as Google software (Google Maps, Street View, and Earth). For more information, see our copyright guidelines: http://journals.plos.org/plosone/s/licenses-and-copyright.

a. You may seek permission from the original copyright holder of Figures 1, 2 and 3 to publish the content specifically under the CC BY 4.0 license. 

Reviewers' comments:

Reviewer's Responses to Questions

**Comments to the Author**

1. Is the manuscript technically sound, and do the data support the conclusions?

Reviewer #1: Yes

Reviewer #2: Yes

2. Has the statistical analysis been performed appropriately and rigorously? 

Reviewer #1: Yes

Reviewer #2: Yes

3. Have the authors made all data underlying the findings in their manuscript fully available?

Reviewer #1: Yes

Reviewer #2: Yes

4. Is the manuscript presented in an intelligible fashion and written in standard English?

Reviewer #1: Yes

Reviewer #2: Yes

5. Review Comments to the Author

Reviewer #1: All-for-One Tourism is an important transformation of the current tourism mode in scenic spots and attractions, endowing tourism with new connotations and marking the entry into a new stage of tourism development. This paper has good innovative significance for choosing this new research theme. The research structure of this article is complete, the data is comprehensive, the research method is appropriate, the research content is rich, and the research conclusion is clear. However, there are two places where modifications are needed: 1. The abstract should supplement the optimization strategies for the development of All-for-One tourism in the Xinjiang Production and Construction Corps of China. 2. All-for-One tourism is a concept proposed by China, which is not an mature international concept. Therefore, before conducting empirical research, it is necessary to introduce the theory of All-for-One tourism.

Reviewer #2: Review Comments:

The manuscript presents innovative advancements in constructing an evaluation system for assessing the development level of all-for-one tourism, building upon the research findings of prior scholars. The authors have crafted an evaluation system for all-for-one tourism that encompasses three dimensions: tourism potential, tourism benefit, and tourism format. This approach renders the evaluation system more comprehensive and aligns well with the current trends of diversified tourism format development.

In the results analysis section, the manuscript exhibits clarity and hierarchy. The authors initiate with an overview of the all-for-one tourism development level within the study area, subsequently delving into a detailed analysis of each region's development level. They further extend their analysis to spatial correlation and coupling coordination from the perspectives of the aforementioned three dimensions, ultimately culminating in the classification of tourist destination types. This analytical trajectory progresses from macro to micro, from general to specific, and from abstract to concrete, providing a robust foundation for subsequent suggestions on all-for-one tourism development.

Additionally, the manuscript holds significance in addressing a research gap. For quite some time, research on all-for-one tourism development has been scant in the specific context of Xinjiang Production and Construction Corps. This manuscript addresses this shortfall to some extent, offering fresh perspectives and serving as a reference for related research fields.

Therefore, I think that this manuscript is novel in concept, rigorous in logic, and thorough in analysis, making it suitable for publication in the PLOS ONE journal. However, to elevate the paper's quality, I recommend the authors consider the following enhancements:

Enhance Readability: Incorporate connecting words and transitional sentences to bolster the logical cohesion between paragraphs, thereby ensuring a tighter structure and facilitating smoother comprehension of your ideas and thesis by readers.

Adhere to Formatting Guidelines: Uniformly modify the paper's layout and format according to the target journal's requirements, encompassing font type, font size, line spacing, margins, and other typographical specifications. This will enhance the paper's professionalism and standardization.

Optimize Figure Clarity: Process the figures by adjusting their resolution, optimizing colors, and so forth, to ensure they are crisp and clear, thereby augmenting their readability and visual appeal.

Include Key Formulas: Insert essential formulas at appropriate junctures, ensuring their accuracy and completeness. This will facilitate the elucidation of your research methodology and elevate the paper's academic rigor and readability.

Expand References: Appropriately incorporate relevant evaluations of all-for-one tourism development levels by past scholars in the reference section to foster comparative analysis with your research. Through comparative argumentation, the innovation and worth of your research can be more effectively demonstrated.

I kindly urge the authors to seriously contemplate and implement these suggestions to further refine the content and elevate the quality of the paper.

6. PLOS authors have the option to publish the peer review history of their article (what does this mean? ). If published, this will include your full peer review and any attached files.

**Do you want your identity to be public for this peer review?** For information about this choice, including consent withdrawal, please see our Privacy Policy .

Reviewer #1: No

Reviewer #2: No

---

## [Author Response · Author response to Decision Letter 1]

19 Dec 2024

Dear Reviewers,

I would like to express my heartfelt gratitude to both of you for your thorough evaluation of our manuscript titled ‘Evaluation of All-for-One Tourism Development Level: Evidence from Xinjiang Production and Construction Corps, China.’ Your insightful comments and constructive suggestions have been invaluable in enhancing the quality of our work. We have carefully considered and meticulously implemented your feedback, and we are pleased to inform you that the manuscript has undergone significant improvements. In the revised version, underlined text indicates language refinements, while yellow-highlighted sections represent newly added content. Below are the detailed responses to your respective comments.

To Reviewer 1:

Abstract Modification: Following your suggestion, we have expanded the abstract to include specific optimization strategies tailored for the development of All-for-One tourism in the Xinjiang Production and Construction Corps of China. These strategies have been carefully chosen to reflect the unique challenges and opportunities faced by this region, thereby enhancing the practical significance and applicability of our research. The inclusion of these strategies provides a clearer picture of our proposed solutions and demonstrates how they can be effectively implemented to foster sustainable tourism growth.

Introduction of the Comprehensive All-for-One Tourism Theory: In recognition of the distinctive nature of the All-for-One tourism concept, we have significantly enhanced our introduction to this theory by incorporating insights from several scholars’ classic conceptualizations. This section not only elucidates the origins, evolution, and core principles of All-for-One tourism but also contextualizes it within the broader framework of China’s tourism development policies. By meticulously reviewing and synthesizing these scholarly contributions, we have ensured that our empirical research is grounded in a robust, coherent, and enriched theoretical framework. This approach allows readers to gain a deeper understanding of All-for-One tourism and appreciate its significance within the context of China's tourism landscape.

To Reviewer 2:

Readability Enhancement: We have carefully reviewed the manuscript and incorporated connecting words and transitional sentences to bolster the logical cohesion between paragraphs. By reducing complex sentence structures and simplifying the language, we have ensured that the ideas flow smoothly and the paper is more accessible to readers. We have also taken care to maintain a consistent tone and style throughout the manuscript, enhancing its readability and overall coherence.

Formatting Compliance: We have thoroughly revised the manuscript to adhere strictly to the formatting guidelines outlined by PLOS ONE. This includes adjustments to font type, font size, line spacing, margins, and other typographical specifications. We have also ensured that the manuscript conforms to the journal's submission guidelines, thereby presenting a professional and standardized appearance.

Figure Clarity Optimization: In response to your suggestion, we have replaced the relevant figures with higher-resolution images and optimized their color schemes for enhanced readability and visual appeal. These images have been carefully validated through PLOS ONE’s image-checking website to ensure compliance with the journal's standards.

Inclusion of Key Formulas: We have added essential formulas at appropriate junctures, accompanied by clear explanations to elucidate our research methodology. This addition not only enhances the academic rigor of the paper but also makes it more accessible to readers who may not be familiar with the technical aspects of our research.

Expansion of References and Discussion: To foster comparative analysis, we have incorporated relevant evaluations of All-for-One tourism development levels by past scholars in the reference section. Furthermore, we have expanded the discussion section to closely integrate and contrast our findings with existing research. This has allowed us to demonstrate the innovation and value of our work within the broader context of the field.

We would also like to express our sincere appreciation for your patience and dedication in reviewing our work. Your insights have been truly enriching, and we are confident that the revised manuscript now presents a more comprehensive, rigorous, and readable contribution to the field.

Thank you once more for your time and effort. Your constructive feedback has been instrumental in refining our research and elevating its quality. We look forward to the opportunity to contribute further to the academic community.

Sincerely,

Chunxiang Zhang

---

## [Decision Letter · Decision Letter 1]

7 Jan 2025

Evaluation of All-for-One Tourism Development Level: Evidence from Xinjiang Production and Construction Corps, China

PONE-D-24-46493R1

Dear Dr. Zhang,

We’re pleased to inform you that your manuscript has been judged scientifically suitable for publication and will be formally accepted for publication once it meets all outstanding technical requirements.

Kind regards,

You-Yu Dai

Academic Editor

PLOS ONE

Additional Editor Comments (optional):

Reviewers' comments:

Reviewer's Responses to Questions

**Comments to the Author**

1. If the authors have adequately addressed your comments raised in a previous round of review and you feel that this manuscript is now acceptable for publication, you may indicate that here to bypass the “Comments to the Author” section, enter your conflict of interest statement in the “Confidential to Editor” section, and submit your "Accept" recommendation.

Reviewer #1: All comments have been addressed

Reviewer #2: (No Response)

2. Is the manuscript technically sound, and do the data support the conclusions?

Reviewer #1: Yes

Reviewer #2: (No Response)

3. Has the statistical analysis been performed appropriately and rigorously? 

Reviewer #1: Yes

Reviewer #2: (No Response)

4. Have the authors made all data underlying the findings in their manuscript fully available?

Reviewer #1: Yes

Reviewer #2: (No Response)

5. Is the manuscript presented in an intelligible fashion and written in standard English?

Reviewer #1: Yes

Reviewer #2: (No Response)

6. Review Comments to the Author

Reviewer #1: (No Response)

Reviewer #2: The author has provided comprehensive responses and revisions to the questions I raised, which have significantly improved the quality of the article. Therefore, I recommend acceptance.

7. PLOS authors have the option to publish the peer review history of their article (what does this mean? ). If published, this will include your full peer review and any attached files.

**Do you want your identity to be public for this peer review?** For information about this choice, including consent withdrawal, please see our Privacy Policy .

Reviewer #1: No

Reviewer #2: No

---

## [Editor Report · Acceptance letter]

PONE-D-24-46493R1

PLOS ONE

Dear Dr. Zhang,

I'm pleased to inform you that your manuscript has been deemed suitable for publication in PLOS ONE. Congratulations! Your manuscript is now being handed over to our production team.

Kind regards,

on behalf of

Dr. You-Yu Dai

Academic Editor

PLOS ONE